# Ice sheets as a missing source of silica to the polar oceans

Jon R. Hawkings[1], Jemma L. Wadham[1], Liane G. Benning[2,3,4], Katharine R. Hendry[5], Martyn Tranter[1], Andrew Tedstone[1,6], Peter Nienow[6] & Rob Raiswell[2]

Ice sheets play a more important role in the global silicon cycle than previously appreciated. Input of dissolved and amorphous particulate silica into natural waters stimulates the growth of diatoms. Here we measure dissolved and amorphous silica in Greenland Ice Sheet meltwaters and icebergs, demonstrating the potential for high ice sheet export. Our dissolved and amorphous silica flux is 0.20 (0.06–0.79) Tmol year$^{-1}$, $\sim$50% of the input from Arctic rivers. Amorphous silica comprises >95% of this flux and is highly soluble in sea water, as indicated by a significant increase in dissolved silica across a fjord salinity gradient. Retreating palaeo ice sheets were therefore likely responsible for high dissolved and amorphous silica fluxes into the ocean during the last deglaciation, reaching values of $\sim$5.5 Tmol year$^{-1}$, similar to the estimated export from palaeo rivers. These elevated silica fluxes may explain high diatom productivity observed during the last glacial–interglacial period.

[1] Bristol Glaciology Centre, School of Geographical Sciences, University of Bristol, University Road, Bristol BS8 1SS, UK. [2] Cohen Biogeochemistry Laboratory, School of Earth and Environment, University of Leeds, Leeds LS2 9JT, UK. [3] German Research Center for Geosciences GFZ, Telegrafenberg, Building C, 14473 Potsdam, Germany. [4] Department of Earth Sciences, Free University of Berlin, 12249 Berlin, Germany. [5] School of Earth Sciences, University of Bristol, Bristol BS8 1RJ, UK. [6] School of Geoscience, University of Edinburgh, Edinburgh EH8 9XP, UK. Correspondence and requests for materials should be addressed to J.R.H. (email: jon.hawkings@bristol.ac.uk).

Silicon (Si) plays a crucial role in global biogeochemical cycles, acting as an essential nutrient for a number of marine organisms, particularly for diatoms, who use it to build their cell frustules and which account for up to 50% of oceanic carbon fixation[1–3]. Diatoms may be especially important in the biological pump and therefore carbon cycle when silica (the most common naturally occuring form of silicon) input is high[4]. Furthermore, chemical weathering of silicate minerals in rock can also sequester atmospheric carbon dioxide on geological timescales[5]. Understanding the components of the Si cycle, constructing silica budgets and evaluating how these have changed in the past and may change in the future is therefore of significant importance.

Over 70% of oceanic dissolved silica (DSi) is derived from riverine input, with the majority of this ($>80\%$) believed to be delivered as DSi[6]. Groundwater, aeolian dust, hydrothermal input and seafloor weathering are also important contributors[6]. Glaciers and ice sheets have largely been neglected in previous DSi budgets, despite covering up to 30% of land surface area during glacial cycles and discharging vast quantities of fresh water and sediment into the coastal ocean[7,8]. There has long been anecdotal evidence linking glacial meltwater and enhanced production of upper and lower trophic species[9,10]. Recent research indicates that glaciers and ice sheets are nutrient factories, delivering substantial fluxes of bioessential nutrients including iron, phosphorus and nitrogen to downstream ecosystems, mainly in reactive particulate form[11–18]. However, the role of glaciers and ice sheets in the global Si cycle has yet to be fully established[8,19], with recent evidence indicating that silica dissolution rates in glacial environments may be higher than previously believed[20,21].

There have been a few studies documenting DSi concentrations in runoff from small valley glaciers[22], but there are currently little data on concentrations or fluxes from large ice sheet catchments. Previous studies have reported low DSi concentrations in glacial meltwater, generally $<30\,\mu M$[22–24], compared with nonglacial rivers (discharge weighted mean of $158\,\mu M$)[25]. We hypothesize that DSi export from large ice sheet catchments may be higher than previously appreciated for two main reasons. First, long water residence times and intense physical erosion rates under large ice sheet catchments[26], and subsequent weathering of fresh mineral surfaces, may promote enhanced silicate dissolution, even with persistently low water temperatures[21,26,27]. Second, as glaciers are also important agents of physical erosion[28,29], meltwaters that emerge from underneath the ice are turbid, and carry fine suspended particulate matter (SPM), often in excess of $1\,g\,l^{-1}$ (ref. 29). Research indicates that the role of terrigenous material (as SPM) in elemental cycles is likely underestimated[30–32]. SPM is likely an important source of DSi because of fine-grained, highly reactive mineral surfaces coated in amorphous nanoparticles[11,12,22]. Glacial SPM needs only to be sparingly soluble to have a large impact on downstream silica and carbon cycling[30,32,33] because of the high sediment load of meltwaters. The impact of glacial SPM on downstream silica budgets has thus far been ignored, and only a small amount of data currently exist[11].

Here we investigate the importance of the Greenland Ice Sheet (GrIS), and by extension former northern hemisphere ice sheets on broadly similar lithologies, for the global Si cycle. We present DSi and easily dissolvable amorphous particulate silica (ASi) concentrations from subglacial meltwaters exciting a glacial catchment in western Greenland. The GrIS provides an accessible ice sheet system, with large, land-terminating glaciers allowing direct sampling of waters of subglacial origin at the ice margin. We highlight the importance of a labile solid-phase amorphous silica phase in meltwaters, with evidence of its dissolution across a glaciated fjord mixing zone. We also document measurements of iceberg-rafted amorphous silica, and calculate potential fluxes from both icebergs and meltwaters discharged from the GrIS to the ocean. We propose that ice sheets deliver a large amount of dissolved and labile amorphous silica downstream. They therefore have a more important role in the silicon cycle, both now and during past glacial-interglacial periods, than previously appreciated.

## Results

**Meltwater and iceberg sampling.** Meltwater samples were collected at least daily during the 2012 melt season from the subglacial channel draining Leverett Glacier, south-west Greenland ($67.1°N\;50.2°W$; Fig. 1). Leverett Glacier is a large outlet glacier of the GrIS ($\sim600\,km^2$ hydraulically active catchment, $\sim80\,km$ long)[29], overlying Archean and Paleoproterozoic igneous shield rock common to much of Greenland, northern Canada and Scandinavia[34]. Iceberg samples were retrieved from a boat in Tunulliarfik Fjord in southern Greenland in 2013 ($61.1°N$, $45.4°W$; Fig. 1), and Sermilik Fjord in eastern Greenland in 2014 ($65.7°N$, $37.9°W$; Fig. 1), using a clean ice axe. These icebergs calved from the floating ice tongues of marine-terminating glaciers nearby. Analytical methods are detailed in the Methods.

**Amorphous and dissolved silica in meltwaters.** Easily dissolvable ASi associated with SPM was the most significant source of potentially bioavailable silica (DSi + ASi) in glacial meltwaters. ASi comprised 0.91 (0.51–1.21) wt.% of SPM equating to very high mean ASi concentrations of 392 (120–627) $\mu M$ (Fig. 2 and Table 1). High-resolution transmission electron microscopic (HR-TEM) and spectral elemental analyses (energy-dispersive spectra (EDS)) confirmed the form of ASi present in meltwaters and icebergs. ASi has not previously been identified and characterized in any natural water SPM using HR-TEM. Amorphous and poorly crystalline nanoparticulate silica was common in all SPM and iceberg debris analysed (Fig. 3). The ASi denudation rate of $>36,000\,kg\,Si\,km^{-2}\,year^{-1}$ (Table 1) exceeds rates found in the literature by at least an order of magnitude, although data are currently sparse.

DSi concentrations in bulk meltwater runoff were generally low, with a discharge weighted mean of $\sim10\,\mu M$ (0.8–41.4 $\mu M$) similar to runoff from other glaciers[23,35]. Higher DSi concentrations ($\sim40\,\mu M$) are at saturation with respect to quartz ($SI_{Qtz} = -0.13$ to $0.09$; Fig. 4), but are still highly undersaturated with respect to amorphous silica ($SI_{ASi} = -1.41$ to $-1.63$). Catchment DSi denudation rates of $\sim980\,kg\,Si\,km^{-2}\,year^{-1}$ are comparable to other glaciers[22] and Arctic rivers[25], despite low DSi concentrations (Table 1).

**Amorphous and dissolved silica in icebergs.** Iceberg ASi concentrations were lower than those found in glacial meltwaters. ASi SPM content at our two sites was 0.27 (0.16–0.46) wt.% for Sermilik fjord icebergs (east Greenland) and 0.28 (0.16–0.47) wt.% for Tunulliarfik fjord icebergs (south Greenland), giving mean ASi concentrations of 47.9 (28.2–81.1) and 50.5 (28.2–83.2) $\mu M$ for Sermilik and Tunulliarfik fjord icebergs, respectively (assuming an estimated sediment load of 0.25–0.75 $g\,l^{-1}$ for icebergs[16]) (Table 1). ASi is concentrated in the sediment-rich, basal ice layers compared with the cleaner ice. Iceberg DSi concentrations were $<20\,\mu M$ in the sediment-rich layers and $<1\,\mu M$ in clean ice layers.

**GrIS dissolved and amorphous silica fluxes.** We estimate GrIS meltwaters currently deliver 0.01 ($<0.01$–0.02) Tmol year$^{-1}$ of

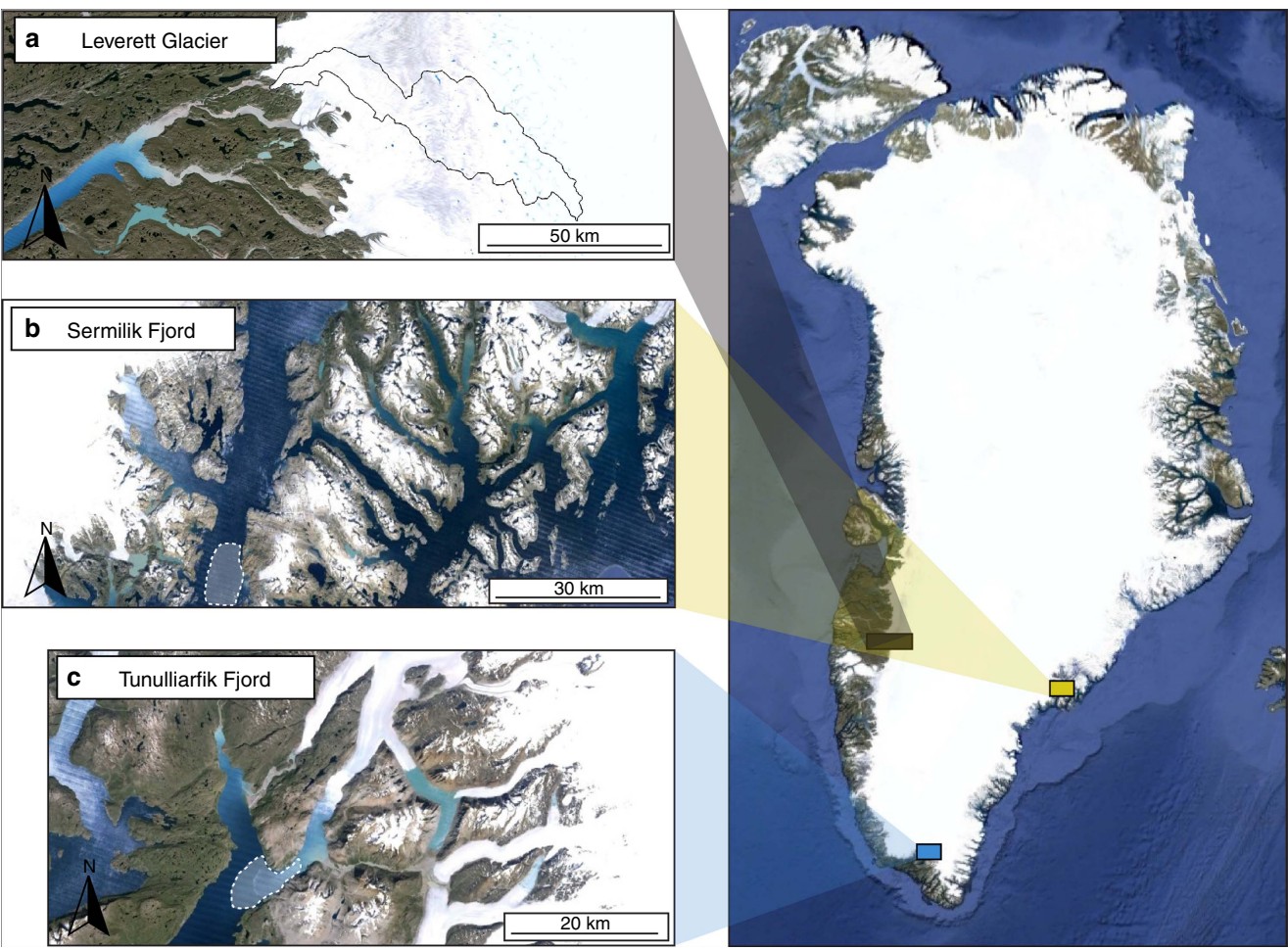

**Figure 1 | Location of study sites in Greenland.** Meltwater samples were collected from (**a**) Leverett Glacier, with the catchment boundary from Palmer et al.[71] outlined. Iceberg samples were collected from (**b**) Sermilik Fjord in east Greenland and (**c**) Tunulliarfik Fjord in south Greenland. The approximate regions where icebergs samples were collected are shaded white in (**b**,**c**). All images are from Google, Landsat, USGS/NASA.

DSi and 0.16 (0.05–0.75) Tmol year$^{-1}$ of ASi to the surrounding fjords and oceans (Table 1), based on the assumption that Leverett Glacier DSi and ASi concentrations are representative of the ice sheet at large. To calculate iceberg fluxes we use the mean iceberg ASi concentration from both sites (0.28%), as ASi concentrations are very similar (Table 1). Icebergs provide an additional flux of 0.03 (0.01–0.04) Tmol year$^{-1}$ of ASi assuming that our samples are representative of other Greenlandic icebergs. Iceberg DSi fluxes are not included because of the low concentrations measured in clean ice ($<1\,\mu$M). These calculations give a total GrIS Si flux of 0.2 (0.06–0.79) Tmol year$^{-1}$. This is 1.8% (0.7–8.4%) of the estimated global Si input to the oceans (9.4 Tmol year$^{-1}$)[6] and nearly 3% (0.9–11.4%) of the terrestrial input (6.9 Tmol year$^{-1}$)[6], despite the GrIS covering only $\sim1.1\%$ of land surface area.

## Discussion

DSi values are an order of magnitude lower than the discharge weighted global mean riverine value of 158 $\mu$M[25] (Table 1), and suggest low silica denudation rates compared with nonglacial river catchments when viewed in isolation. DSi concentrations, $SI_{Qtz}$ and $SI_{ASi}$ track the evolution of subglacial drainage (Figs 2 and 4). Higher concentrations ($\sim40\,\mu$M) were found at the onset of the melt season, when an inefficient drainage system was present at the glacier bed and dilution by dilute supraglacial meltwater was lower[36]. As the melt season progressed, lower

DSi concentrations and electrical conductivity, as efficient drainage pathways opened up, suggest greater dilution by supraglacial meltwater (Fig. 2), with waters becoming increasingly undersaturated ($SI_{Qtz}$ and $SI_{ASi}$ minimums are $-1.71$ and $-3.21$, respectively).

Glacial silica export is dominated by the ASi fraction (Table 1). The ASi fraction of SPM (0.51–1.21 wt.%) was comparable to those measured in the Ganges basin (mean 1.2% by weight)[37], which is characterized by high sediment yields[38], and is higher than the estimated global river SPM ASi of 0.6 wt.%[33]. The corresponding mean concentrations of ASi in glacial meltwaters were nearly six times higher than the concentrations measured in the Ganges basin (68 $\mu$M)[37] and far exceed the mean concentration of ASi in river waters given by Conley[39] of 28 $\mu$M that is used in recent ocean silica budget estimates[6]. This difference is caused by both higher mean SPM concentrations ($\sim1$ versus $\sim0.1$ g l$^{-1}$)[38] and the higher mean ASi ($\sim0.9$ versus $\sim0.6$ wt.%)[33] in glacial runoff. ASi was mostly associated with the fringes of larger platy material (Fig. 3), suggesting it is a product of aluminosilicate mineral weathering[40] and/or mechanical grinding[41,42]. EDS of most ASi identified the incorporation of other elements into the ASi nanostructures, most commonly Al and Fe (Fig. 3a,b). This is not unexpected as naturally occurring mineral ASi incorporates less soluble elements as impurities during formation, because of its loose structure and high water content[43].

It has previously been demonstrated that iceberg-rafted debris is likely a large source of reactive nanoparticulate iron to the euphotic zone[16]. Our results indicate that icebergs also have the capacity to supply ASi to surface waters. Iceberg ASi concentrations (Table 1) fall at the lower end of concentrations reported for glacial and some nonglacial waters, but exceed

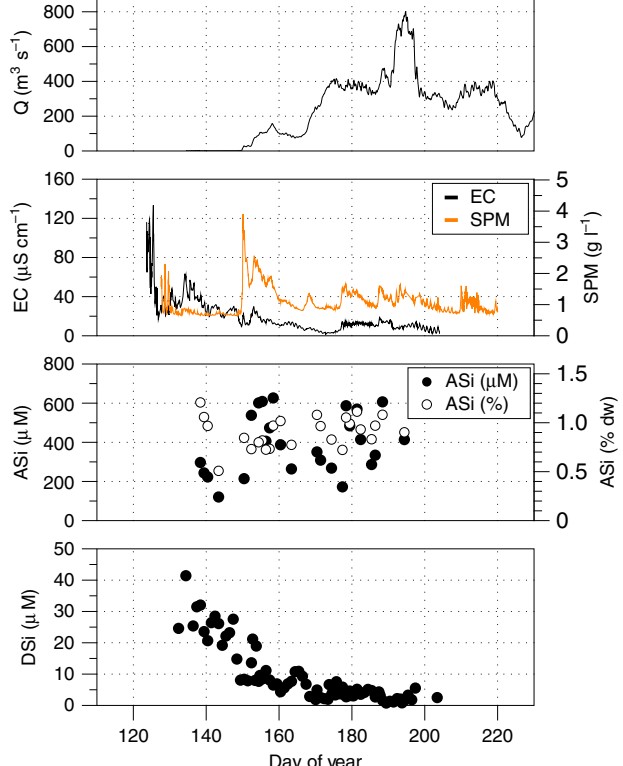

**Figure 2 | Time series of hydrological and dissolved/amorphous silica data from the proglacial river of Leverett Glacier.** (**a**) Leverett Glacier discharge (Q), (**b**) meltwater electrical conductivity (EC) and suspended particulate matter (SPM), (**c**) meltwater amorphous silica (ASi) concentration in $\mu M$ and percentage dry weight of sediment (% dw), and (**d**) meltwater dissolved silica (DSi) concentration.

the Conley[39] estimated concentration of ASi in river waters. Significantly lower concentrations of ASi in iceberg-rafted debris compared with our glacial meltwaters reflect unsorted sediments in iceberg sediment bands versus finer more reactive material carried as SPM in glacial meltwaters, and lower SPM concentrations in icebergs versus meltwaters ($\sim 0.5$ versus $\sim 1\,\mathrm{g\,l^{-1}}$).

Recent studies have exploited the use of modern nano-observation technologies to study ASi formation on mineral surfaces from aqueous weathering processes[40]. Two chemical weathering mechanisms have previously been identified. First, the dissolution–reprecipitation mechanism, where ASi forms as a precipitated weathering crust on freshly ground and leached particles. This has been observed even in solutions that are significantly undersaturated with respect to silica. Second, the leached surface layer hypothesis[44], where preferential removal of weakly bonded ions (for example, $Na^+$ and $K^+$) from the mineral surface leave an amorphous crust rich in more insoluble ions such as silica. Both mechanisms invoke higher chemical weathering rates in subglacial environments than previously realized[20], as ASi concentrations are high.

The comminution of bedrock by glaciers and ice sheets is also likely to be important in producing structural change to mineral surfaces[41,42,45]. Grinding of quartz produces a disturbed amorphous surface layer[42] that is much more soluble than the primary mineral[41]. For example, Henderson et al.[41] found freshly ground silica particles were more than an order of magnitude more soluble (115 p.p.m.) than 'cleaned' crushed quartz particles (11 p.p.m.) at pH 8. Silicate minerals that have been freshly abraded by glacial action are therefore likely to be substantially more soluble than unaltered mineral surfaces.

There is considerable uncertainty around the lability of ASi before long-term burial in fjords and near coastal regions. We believe ASi associated with glacial sediments will be highly labile downstream for three main reasons. First, glacial rock flour is potentially highly reactive because of a disturbed surface layer and large surface area per unit mass[35,45], and because it has been observed to form buoyant flocs on contact with salt water[46]. Second, the extraction protocol used is designed to capture the silica that will likely dissolve in sea water (that is, the highly labile component)[33,47]. Last, ASi (and unreactive silicate minerals) dissolution is catalysed by the presence of alkali metals (for

**Table 1 | Mean silica concentrations, yields and estimated fluxes for global rivers, Pan-Arctic rivers and the Greenland Ice Sheet.**

| | Global rivers | Pan-Arctic rivers | Greenland Ice Sheet | | Antarctic Ice Sheet | |
| --- | --- | --- | --- | --- | --- | --- |
| | | | Meltwater | Icebergs | Meltwater | Icebergs |
| DSi ($\mu M$) | 158 (ref. 25) | 102 (ref. 25) | 9.6 (0.8–41.4) | — | 130–210 (ref. 27) | — |
| ASi (% dry weight) | 0.6 (ref. 33) | 1.2*,(ref. 37) | 0.91 (0.51–1.21) | 0.28 (0.16–0.47) | — | — |
| ASi ($\mu M$) | 28 (ref. 39) | 24† | 392 (120–627) | 49 (28–83) | — | — |
| Total discharge ($km^3\,year^{-1}$) | 39,080 (ref. 79) | 3,310 (ref. 25) | 437‡,(ref. 77) | 612§ (ref. 58) | 65 (ref. 63) | 1,321 (ref. 64) |
| Total SPM load (Tg) | 12,800 (ref. 80) | 207 (ref. 25) | 485 (300–1,700) | 306 (150–459) | — | — |
| DSi yield ($kg\,Si\,km^{-2}\,year^{-1}$) | 1,500 (ref. 25) | 560 (ref. 25) | 980‖ | — | — | — |
| ASi yield ($kg\,Si\,km^{-2}\,year^{-1}$) | 830¶,(ref. 33) | 160† | 36,000‖ | —# | — | — |
| DSi flux (Tmol) | 6.2 ± 1.8**,(ref. 6) | 0.34 (ref. 25) | 0.01 (0–0.02) | — | 0.01 | — |
| ASi flux (Tmol) | 1.1 ± 0.2 (ref. 6) | 0.09 | 0.16 (0.05–0.75) | 0.03 (0.01–0.04) | 0.01†† | 0.06‡‡ |
| % Of global budget | 78 | 4.6 | 1.8 | 0.4 | 0.2 | 0.6 |

*Unknown and hence upper limit taken from Frings et al.[37]
†Calculated from mean SPM of $0.06\,g\,l^{-1}$ for Arctic rivers and extractable SPM ASi of 1.2%.
‡Mean meltwater discharge from 2000 to 2012.[77]
§Mean solid ice discharge from 2000 to 2010.[58]
‖Based on Leverett Glacier catchment area and corresponding catchment DSi/ASi flux.
¶Assuming mean riverine ASi of 0.6% from Frings et al.[33].
#Catchment data unavailable.
**Does not include reduction ($\sim 25$%) due of reverse weathering and trapping in the estuary.
††Estimated using the lowest Leverett Glacier ASi concentration (120 $\mu M$).
‡‡Estimated using mean Greenland iceberg ASi concentration (49 $\mu M$).

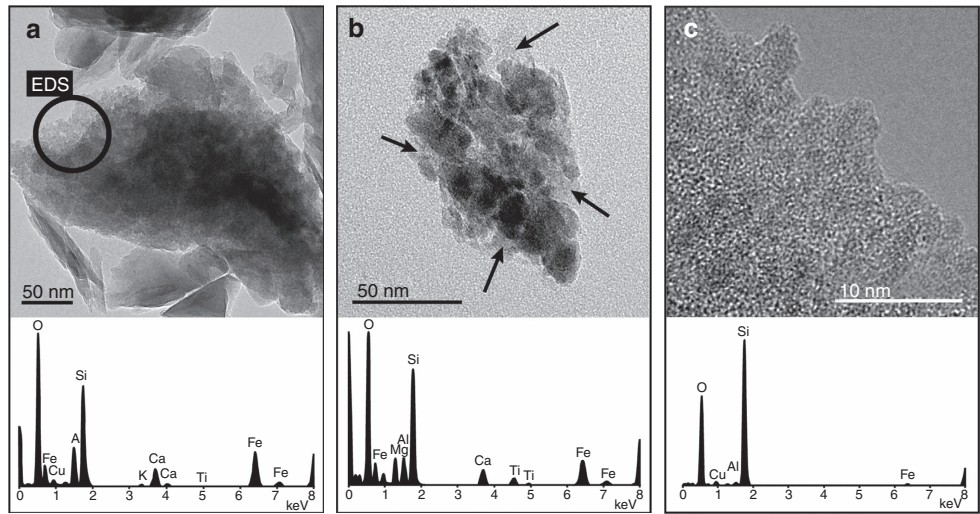

**Figure 3 | Photomicrographs of glacial sediments.** Representative form of amorphous silica (ASi) identified in (**a**) Leverett Glacier suspended particulate matter, (**b**) Tunulliarfik and (**c**) Sermilik iceberg-entrained sediment. Energy-dispersive spectra (EDS) labelled circle in (**a**) indicates region where EDS spectra were acquired. EDS spectra in (**b**) were acquired from the whole particle, and may therefore include some aluminosilicate material (with high Al content). EDS spectra in (**c**) were of an enlarged area characterized by poorly ordered ASi nanoparticles. Arrows in (**b**) indicate regions where ASi nanoparticles were observed similar to those in (**a**,**c**).

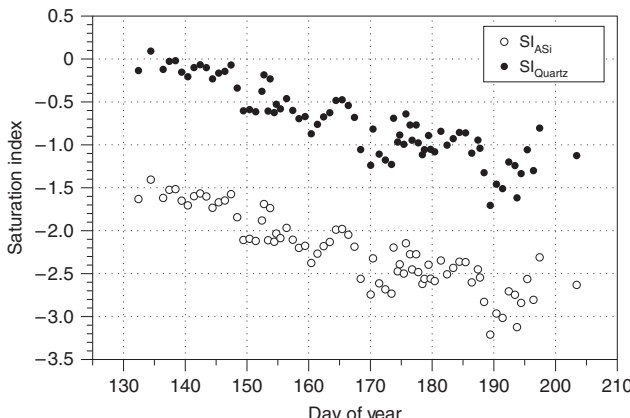

**Figure 4 | Saturation index of quartz and amorphous silica in Leverett Glacier meltwaters over the 2012 season.** Saturation index of amorphous silica ($SI_{ASi}$) and quartz ($SI_{Quartz}$) were calculated using the Geochemists Workbench software package with our hydrogeochemical data set.

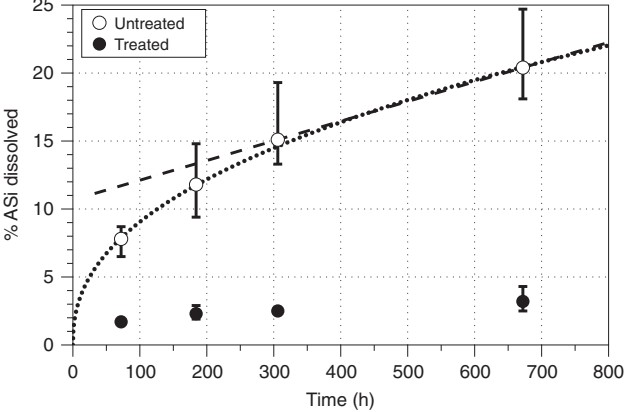

**Figure 5 | Percentage dissolution of amorphous silica from Leverett Glacier suspended particulate matter in low Si seawater leach.** Points indicate the mean of four replicate leaches, with bars showing the minimum and maximum values attained. The dashed and dotted lines show the dissolution fits used to estimate complete amorphous silica (ASi) dissolution time. The percentage total ASi in sediments used in the seawater leach was calculated in triplicate extractions using the 0.1 M $Na_2CO_3$ extraction, documented in the Methods.

example, $Na^+$) and alkali earth metals (for example, $Ca^{2+}$)[48]. ASi is therefore expected to dissolve much more rapidly in marine waters than fresh waters[49,50], generating DSi for diatom uptake. ASi has been found to be up to two orders of magnitude more soluble in saline waters than fresh waters[49,51] and is expected to dissolve relatively rapidly (on timescales of days to weeks)[50]. For example, Kato and Kitano[50] found complete dissolution of 50 mg of synthetic ASi in 1 litre of artificial sea water in <22 days.

We performed a simple seawater leach on Leverett Glacier SPM to determine the lability and therefore potential bio-availability of ASi (see Methods for details). This demonstrated rapid release of DSi from ASi over a period of days to weeks (Fig. 5). Treated sediment, with ASi removed before leaching (pre-extracted with 0.1 M $Na_2CO_3$), showed only minor Si dissolution over a period of 672 h (28 days; Fig. 5) compared with untreated sediment. Untreated sediments displayed up to 25% ASi dissolution over the same time period, indicating ASi will likely dissolve relatively rapidly in high salinity waters. The DSi

(measured as silicic acid) released into solution is bioavailable to marine diatoms. We propose two possible scenarios for longer-term dissolution of ASi in saline waters (>28 days). The first uses a linear dissolution function derived from the final two time points (306 and 672 h; dashed black line in Fig. 5). Under this scenario complete ASi dissolution would occur within 259 days (~9 months). The second uses a more conservative power fit function derived from all time points (dotted black line in Fig. 5). Under this scenario, at least 60% of ASi dissolves within a year. We therefore hypothesize that 60 to 100% of SPM ASi will dissolve within a year in saline waters. Benthic processing of glacial material, and delivery back into the euphotic zone, is likely to be important on longer timescales, as has been demonstrated in other fjord environments[52].

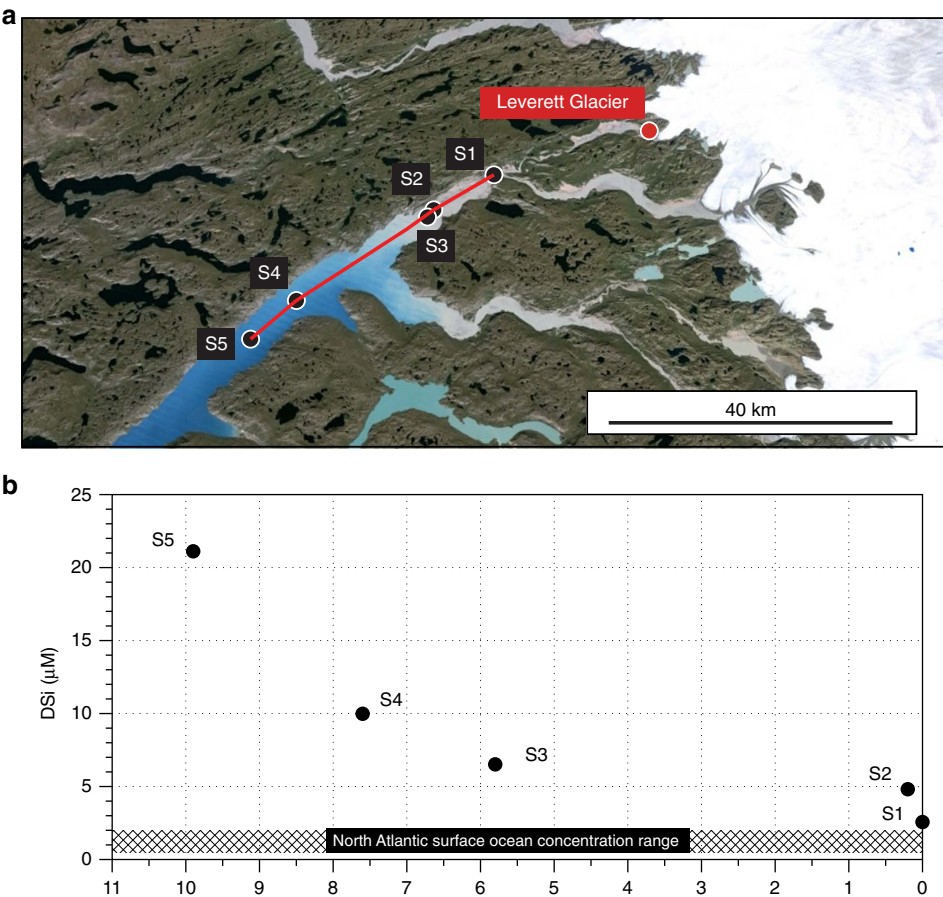

**Figure 6 | Søndre Strømfjord transect of surface dissolved silica concentrations.** (**a**) Satellite image of the Leverett Glacier study region. The position of Leverett Glacier terminus, at the head of Watson River, is given. S1 indicates the point at which Watson River exits the settlement of Kangerlussuaq. S2–S5 indicate sampling points along Søndre Strømfjord. (**b**) Concentrations of dissolved silica (DSi) plotted against salinity at sampling points S1–S5. The shaded region indicates the approximate range of regional sea surface dissolved silica concentrations from Painter *et al.*[54]. The plot *x* axis is reversed to reflect site positioning in (**a**). The satellite image in (**a**) is from Google, Landsat, USGS/NASA.

We found further evidence of rapid ASi dissolution from Greenlandic meltwaters, with more than an order of magnitude increase in DSi concentrations across a buoyant SPM-rich glacial meltwater plume mixing with saline fjord waters downstream of Leverett Glacier (Fig. 6)[53]. Our data set represents a limited number of observations and a snapshot in time, but the positive association between DSi and salinity is contrary to what is usually observed in nonglacial estuaries and deltas, where there is removal of ∼25% DSi because of reverse weathering and diatom uptake[6]. The DSi concentration observed at S5 (21.1 μM; ∼40 km from S1) is an order of magnitude greater than oceanic surface water DSi concentrations in the North Atlantic (generally <2 μM), despite the high diatom productivity observed in west Greenland fjords[19,53]. Recent studies have also recorded higher concentrations (mean concentration of 2.22 μM) of surface DSi in coastal and open ocean waters on the Greenland Shelf[54] compared with the North Atlantic. In a similar manner, iceberg ASi will also likely provide an important source of DSi as iceberg-rafted debris melts out in marine waters. Enhanced primary production has been recorded in the wake of icebergs, through observation of surface chlorophyll concentrations[55,56], and diatom communities have been observed growing on the underside of icebergs in the Southern Ocean[57]. This is consistent with icebergs being a primary source of nutrients, including silica, to ocean surface waters.

The glacial impact on the marine Si cycle and associated budgets will depend on the magnitude of the glacial flux and the lability of the exported Si. High rates of physical weathering[29] and the presence of a labile ASi solid-phase indicative of subglacial silicate mineral chemical and/or physical weathering mean that ice sheets are likely a significant source of DSi to downstream fjords and near coastal regions. Concentrations derived from Leverett Glacier are likely to be typical of other large land-terminating outlet glaciers that export large quantities of meltwater from the GrIS following drainage across the glacier bed[58]. There are clear limitations to using a single glacier to estimate Si meltwater export from the GrIS and we acknowledge there may be large uncertainties in our estimates because of the extrapolations we have made. However, Leverett Glacier is significantly larger (by almost two orders of magnitude) than any other glaciated catchment reported thus far in the literature (both in Greenland and worldwide). The underlying geology[59] and catchment hydrology[60] are likely typical of other large land-terminating outlet catchments of the GrIS and therefore the values we derive are a reasonable first-order approximation of GrIS fluxes, until more data become available.

GrIS dissolved and amorphous silica fluxes are comparable to the total estimated input from atmospheric deposition (0.5 Tmol year$^{-1}$), groundwater (0.6 Tmol year$^{-1}$) and hydrothermal sources (0.6 Tmol year$^{-1}$)[6]. We find it is likely to be the most dominant single source of dissolved and amorphous silica to

the pan-Arctic region if we compare the GrIS Si flux with Arctic rivers. Using DSi estimates from Durr et al.[25], and an upper ASi of ~1.2 wt.% for riverine SPM (an upper estimate derived from Frings and co-workers[33,37] as there are no data for Arctic rivers), we estimate an Arctic riverine Si input of 0.35 Tmol year$^{-1}$ (Table 1). The GrIS could therefore provide up to ~37% of total DSi + ASi input into the coastal regions of Arctic seas >60°N (~50% of the total nonglacial riverine flux). The wider impact of these fluxes will depend on physical oceanographic factors around the GrIS that may not favour significant off-shelf export[61]. Processing of dissolved and amorphous silica may also limit the flux of silica out of long fjord systems. However, glaciated fjords harbour highly productive microbial ecosystems[53], are important feeding grounds for seabird and marine mammals[9] and have been identified as regions of high carbon burial[62].

The Antarctic Ice Sheet (AIS) may also be a significant source of dissolved and amorphous silica to the Southern Ocean. Previous published estimates indicate the AIS DSi flux is in the region of ~0.1 Tmol year$^{-1}$ (ref. 6), but it neglected the potential export of ASi attached to SPM and iceberg-hosted sediments. We make a comparison with these original estimates using results from more recent research combined with our data of GrIS ASi concentrations to provide a revised approximation of the silica flux from the AIS. The only meltwaters to be sampled from the basal environment of the AIS come from subglacial Lake Whillans[27]. These waters indicate that Antarctic meltwaters are enriched in DSi compared with GrIS meltwaters, likely because of the long residence time of waters, and lack of dilution by incoming supraglacial melt (as in the GrIS). This study suggests that DSi concentrations in subglacial Antarctic meltwaters may be between 130 and 210 µM[27], similar to the mean nonglacial global riverine estimate[25]. We estimated the DSi contribution from AIS subglacial meltwater using modelled basal melt rates of 65 km$^3$ year$^{-1}$ (ref. 63). This gives a meltwater DSi flux of ~0.01 Tmol year$^{-1}$, similar to the GrIS DSi flux (Table 1). AIS meltwater sediment flux is highly uncertain as no measurements exist. We therefore use a conservative ASi concentration estimate of 120 µM (the lowest value recorded at Leverett Glacier) with the above meltwater flux[63]. This gives a total AIS DSi + ASi meltwater flux of ~0.02 Tmol year$^{-1}$ that is a similar order of magnitude to the flux of Treguer[8] (0.04 Tmol year$^{-1}$), but substantially less than the GrIS (0.2 Tmol year$^{-1}$). Iceberg calving fluxes are significantly higher from the AIS than the GrIS. Depoorter et al.[64] estimate an iceberg calving flux of 1,321 ± 144 km$^3$ year$^{-1}$ from the AIS. If we assume a similar sediment loading (0.5 g l$^{-1}$) and ASi wt.% to our GrIS iceberg estimates (Table 1), this gives a AIS iceberg flux of ~0.06 Tmol year$^{-1}$. Our estimated AIS dissolved and amorphous silica flux is therefore in the region of ~0.08 Tmol year$^{-1}$, around half that of the GrIS, and similar to the previous AIS estimate (~0.1 Tmol year$^{-1}$)[8]. We estimate that the total ASi + DSi flux from the AIS and GrIS is therefore ~0.3 Tmol year$^{-1}$, ~3% of the global Si budget (Table 1). However, the AIS DSi and ASi flux estimate remains speculative because of uncertainties in subglacial meltwater discharge and DSi concentrations, as well as no data on ASi concentrations for SPM in AIS meltwaters or iceberg-hosted sediments.

Studies postulate a link between the supply of Si to the ocean and the efficiency of the biological carbon pump[1–3]. Diatoms dominate the phytoplankton community during periods where the silica flux to the oceans is high, and are likely more efficient exporters of carbon than other primary producers[2]. Peaks in diatom abundance in marine sediment records from the last deglaciation have previously been explained by enhanced surface supply of DSi as a result of changes in ocean circulation and upwelling[65–68]. However, here we suggest that glacial runoff and

iceberg-entrained debris may deliver an additional high DSi + ASi flux during deglaciation, especially during meltwater pulse events and Heinrich events. We construct crude estimates of palaeo ice sheet fluxes of DSi and ASi to the oceans using recent model estimates for meltwater release during the last deglaciation[69]. These calculations indicate that meltwater pulse event 1a (~15,000 to 14,500 years before present) contributed meltwater discharge of at least 15,000 km$^3$ year$^{-1}$, equivalent to sea level rise of >4 cm year$^{-1}$. A crude calculation indicates ice sheets would have delivered on the order of 5.7 Tmol year$^{-1}$ of DSi + ASi to the oceans, assuming a similar SPM, ASi and DSi concentration to modern-day Leverett Glacier (Table 1). Nonglacial riverine discharge was likely significantly lower during the Last Glacial Maximum compared with present day (by at least 20–25%)[70]. Our estimated palaeo ice sheets flux is therefore similar to the approximate DSi + ASi flux for palaeo rivers (~5.5–5.8 Tmol year$^{-1}$, assuming nonglacial riverine silica fluxes broadly scale with discharge). The impact of the palaeo ice sheet Si flux will be felt for an extended period after input[1], given the long residence time of Si in the oceans of >10,000 years[33].

Our findings indicate that ice sheets play a more significant role in the global Si cycle than previously recognized, mainly via export of large quantities of potentially labile amorphous silica. This phase dominates the glacial dissolved and amorphous silica meltwater flux, with ASi concentrations up to 627 µM and yields of >36,000 kg Si km$^{-2}$ measured at a large ice sheet catchment. Our flux estimates of dissolved and amorphous silica for the GrIS demonstrate that meltwater and iceberg discharge are significant and may provide similar amounts to the oceans as dust deposition, groundwater discharge and hydrothermal input. Hence, the GrIS likely contributes a large proportion of the dissolvable silica in the productive fjord and near coastal regions, where diatoms make up a large proportion of the phytoplankton community. These results indicate that glaciated regions play a more important role in the Si cycle than previously appreciated, and should be considered in future marine dissolved and amorphous silica budgets. Our findings have significant implications for the understanding of the Si cycle in the past, with globally significant fluxes of silica into the oceans likely during catastrophic melting of the large palaeo ice sheets that covered nearly 30% of land surface area. Large ice sheet pulses of dissolved and amorphous silica during these periods are a viable driver of deglacial diatom-dominated phytoplankton communities as observed in core records, in turn potentially enhancing the efficiency of the biological pump.

## Methods

**Study areas.** Glacial meltwater samples were collected from Leverett Glacier (67.1°N, 50.2°W) in 2012. Leverett Glacier is a large land-terminating outlet of the GrIS. It is ~80 km long, and has a hydrologically active catchment area of ~600 km$^2$ (refs 29,71). Mean summer discharge in 2012 was >200 m$^3$ s$^{-1}$ (ref. 12). Runoff feeds a large glacial river system, Watson River, that discharges into Søndre Strømfjord. The glacier overlies predominantly Precambrian crystalline bedrock, typical of large areas of Greenland[59]. The catchment hydrology is well documented and although comparative data sets are thus far lacking, it is believed typical of the large Greenland outlet glaciers that dominate discharge of meltwaters from the GrIS[36,72]. In addition, fjord samples were taken from a 30 km transect of Søndre Strømfjord, downstream of Leverett Glacier, in 2012 (Fig. 3).

Iceberg samples were collected from Tunulliarfik Fjord (61.1°N 45.4°W) in July 2013 and Sermilik Fjord (65.7°N 37.9°W) in July 2014. Both fjords receive ice discharged from local marine-terminating glaciers and were sampled at least ~18 km (Tunulliarfik Fjord) and ~40 km (Sermilik Fjord) downstream of where they calved. The shield bedrock geology from these catchments is broadly similar to Leverett Glacier [59].

**Sample collection and filtration.** Bulk meltwater samples were collected at least once daily (1,000–1,200 h and occasionally 1,800–2,000 h) throughout the main melt period (May, June, July for Leverett Glacier in 2012)[11,18]. Grab samples were collected in 2l high-density polyethylene (HDPE) Nalgene bottles rinsed three

times before final sample collection. Meltwater samples for DSi were filtered through a 47 mm 0.45 μm cellulose nitrate filter (Whatman), mounted onto a PES filter Unit (Nalgene) at Leverett Glacier. Three replicate samples collected at Leverett Glacier using the filter unit and syringe filter methods showed no significant difference in final measured concentration (± 2%). DSi samples were stored in clean in the dark 30 ml HDPE bottles (Nalgene) rinsed three times with filtrate, refrigerated to prevent polymerization and analysed within 3 months of collection. Samples for ASi ($n = 25$) were collected from the retained sediment on the cellulose nitrate filter. These were stored air dried and refrigerated until analysis.

Fjord water was collected (17 June 2012) using a 0.45 μm Whatman GD/XP PES syringe filter using a PP/PE syringe. Surface water samples were collected in 2 L HDPE bottle. Bottles were rinsed three times with sample water and then fully immersed ∼0.3 m below the surface to collect the final sample. Salinity and pH were taken at each sampling site.

Iceberg samples were retrieved from a boat in Tunulliarfik Fjord ($n = 12$), southern Greenland and Sermilik fjord ($n = 5$), eastern Greenland, using a clean ice axe. Excavated blocks of iceberg were placed in new, clean Whirl-Pak bags. The outer layer of ice was allowed to melt and was discarded to minimize potential contamination from the sampling process. The remaining ice was transferred to a new Whirl-Pak bag and allowed to melt completely. Iceberg-entrained sediments were collected by filtration of the melted ice through a 47 mm 0.4 μm Whatman Cyclopore PC membrane filter mounted on a Nalgene PS filtration unit. The filtrate was retained for analysis of DSi.

**Analytical procedures.** DSi was determined using a LaChat QuikChem 8500 series 2 flow injection analyser (QuikChem Method 31-114-27-1-D). This method uses the well-established molybdic acid colourimetric method. Seven standards (matrix matched for fjord samples) were used, ranging from 10 to 2,000 μg l⁻¹ Si (0.36–71.43 μM). The methodological limit of detection was 0.3 μM, precision ± 0.5% and accuracy −1.2%.

ASi was determined using an alkaline digestion[47]. This weak base digestion is believed to dissolve amorphous/poorly crystalline silica. It is commonly employed to determine biogenic opal and pedogenic opal in marine waters, as well as adsorbed Si and poorly crystalline aluminosilicates in terrestrial soils and sediments[47,73]. Approximately 30 mg of sediment was accurately weighed into a 60 ml HDPE bottle (Nalgene), with 50 ml of a 0.096 M $Na_2CO_3$ solution added. Bottles were placed in an 85 °C hot water bath, and 1 ml aliquots of samples were taken from the sample bottle after 2, 3 and 5 h, using a precalibrated 1 ml automatic pipette. Aliquots were stored in 2 ml PP microcentrifuge tubes at 4 °C until analysis < 24 h later. Before analysis, 0.5 ml of sample was neutralized with 4.5 ml of 0.021 M HCl in a 10 ml plastic centrifuge tube. Samples were analysed using the dissolved silica method described above. Total ASi was determined by calculating the intercept of a linear regression line through collected aliquots at 2, 3 and 5 h, assuming amorphous Si phases dissolve completely within the first hour of extraction and clays/more crystalline material release DSi at a constant rate over the experimental time frame[47,74,75].

ASi is expressed as both % dry weight and as concentration in μM. The ASi wt.% was calculated using the weight of sediment added for each extraction, the amount of extraction solution at each sampling time point (2, 3 and 5 h) and the amount of dissolved silica in the solution at that time point. The final ASi % was the value of intercept of the linear regression as described above. The concentration of ASi was derived from the SPM concentration (in g l⁻¹) at the sampling time point and the wt.% ASi of that sample. The ASi concentration was then converted from mg l⁻¹ to μM.

**Seawater sediment leach.** A simple leach on SPM from Leverett Glacier was conducted using natural low Si sea water. The seawater DSi concentration was 1.6 μM, and the salinity 35.49 PSU. All sea water was sterilized by filtration through a 0.22 μm PES Stericup filtration unit. Two types of Leverett Glacier SPM were used. Both were derived from the same bulk SPM sample (itself an amalgamation of SPM from several time points to create a homogenous sample). The first, 'treated' sediment, was extracted using the 0.1 M $Na_2CO_3$ procedure described above, before the seawater leach to remove SPM-associated ASi. The second, 'untreated', was SPM with no prior treatment (that is, natural). Four 15 ml centrifuge tubes (PET, Fisherbrand) were filled with 10–15 mg of accurately weighted sediment and 10 ml of sea water (1–1.5 g l⁻¹) for each of the four time points (72, 184, 306 and 672 h) when measurements were taken. Tubes were incubated at lab temperature (18 °C) in the dark, and gently agitated on an orbital shaker (at 50 r.p.m.). Leaches were terminated by filtration of 5 ml aliquots through a 0.22 μm syringe filter (PES, Millex) into a clean 15 ml centrifuge tube. Aliquots were stored refrigerated before analyses for DSi within 24 h of sampling, following the procedure above. Results are expressed as the percentage of ASi that has dissolved. The ASi content of the all sediment used was determined according to the analytical procedure above, to calculate this.

**Micro-spectroscopic analysis.** The morphology and structure of silica phases was determined using high-resolution field emission gun transmission electron microscopy (HR-TEM; FEI Tecnai TF20) operating at 200 kV. Particulate material

were removed from the filters and dispersed in ethanol using an ultrasonic bath for ∼1 min. A drop of this solution was pipetted onto a standard holey carbon support cupper grid. High-resolution images were complemented by (acquired with an Oxford Instrument analyses system) to determine the elemental characteristics of material. Amorphous silica was characterized by the lack of any crystallinity and the large Si and O peaks in the EDS, although other cations were also often present.

**Hydrological monitoring and mass fluxes.** Leverett Glacier was hydrologically gauged at stable bedrock sections throughout the 2012 ablation season from late April until mid-August, according to the methods detailed by others[11,72,76]. Stage was converted to discharge using ratings curves determined from rhodamine dye-dilution experiments ($n = 41$) over a range of water levels. Errors associated with discharge readings are estimated to be < 15% (ref. 76). Total meltwater fluxes were determined from the cumulative sum of discharge over the ablation season.

Suspended sediment concentrations were determined using a season-long record of turbidity. This turbidity record was converted to suspended sediment concentrations by calibration with manually collected samples; 300–500 ml of meltwater was filtered through a pre-weighed 47 mm 0.45 μm cellulose nitrate filters (Whatman). Filters were subsequently oven dried overnight and re-weighed. Uncertainty in suspended sediment measurements is estimated to be ± 6% (ref. 29).

**Flux estimates.** GrIS freshwater fluxes were derived using mean modelled meltwater runoff from 2000 to 2012 of 437 km³ year⁻¹ from Tedesco et al.[77] Minimum, mean and maximum sediment fluxes are derived from minimum, discharge weighted mean and maximum suspended sediment concentrations measured at Leverett Glacier. This produces a total sediment flux of 300–1,700 Tg year⁻¹. An iceberg mass flux of 612 km³ year⁻¹ (2000–2010) is derived from the solid ice discharge data of Bamber et al.[58] We use an estimated minimum, median and maximum iceberg sediment load of 0.25, 0.5 and 0.75 g l⁻¹ as sediment content of icebergs is poorly constrained. These values may be conservative as higher concentrations of 0.6–1.2 g l⁻¹, based on excess ²²⁴Ra activity in the vicinity of icebergs in the Weddell Sea, have been estimated by others[78]. This produces a GrIS iceberg sediment flux of 150–459 Tg year⁻¹.

Paleo ice sheet fluxes were calculated using modelled ice sheet reconstruction data (ICE-6G_C) from Peltier et al.[69] These meltwater flux values are likely underestimates as they assume no accumulation on ice sheets during the period of estimated mass loss; for example, the net mass balance may be −15,000 km³ year⁻¹ (contribution to sea level rise), but glacial runoff may have been ∼25,000 km³ year⁻¹ with ∼10,000 km³ year⁻¹ of mass accumulation in the ice sheet interior.

**Data availability.** The data used in this article are available from the corresponding author (jon.hawkings@bristol.ac.uk) on request.

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

## Acknowledgements

This research is part of the UK NERC funded DELVE (NERC grant NE/I008845/1) and associated NERC PhD studentship to J.R.H., the UK NERC Arctic Soils grant (NERC grant NE/J022365/1) and the German Helmholtz Recruiting Initiative to L.G.B. A.T. was funded by a NERC studentship and MOSS scholarship. P.N. was supported by grants from the Carnegie Trust for University of Scotland and The University of Edinburgh Development Trust. The Leverhulme Trust, via a Leverhulme research fellowship to J.L.W., provided additional support. We thank all those who assisted with fieldwork at Leverett Glacier, as well as Mr James Williams and Dr Fotis Sgouridis in LOWTEX Laboratories at the University of Bristol, and Dr Mike Ward at the LEMAS facility in the University of Leeds. We finally thank Dr Paul Carrow for his input and advise on the seawater leaching experiments. We are grateful to our anonymous reviewers for their constructive comments on the manuscript.

## Author contributions

All authors made significant contributions to the research presented here. J.R.H., J.L.W. and M.T. conceived the project. J.R.H., R.R., A.T. and P.N. collected the field data. J.R.H., L.G.B. and K.R.H. undertook lab analysis. J.R.H. wrote the manuscript with contributions from all authors.

## Additional information

**Competing financial interests:** The authors declare no competing financial interests.

