## [Peer Review File · Nature Communications]

Reviewers' comments:

Reviewer #1 (Remarks to the Author):

A and B-Key results/Originality/interest:

Silicon is one of the most important element in marine biogeochemistry. This element is absolutely required for the growth of diatoms which are key players of the biologically driven carbon pump. The external sources of silicic acid (DSi) to the oceans derive from the weathering of the Earth's crust. The contribution of ice sheets has been neglected so far.

This study is a new contribution to push up the idea that glaciers and ice sheets are nutrient factories.

From data related to the discharge of the Leverett Glacier and to icebergs floating in fjords of the southern and eastern Greenland, this manuscript shows that ice sheets melt waters are likely to be a significant source of DSi and especially of easily dissolvable amorphous silica (ASi) to downstream fjords and coastal regions.

Extrapolation to the Greenland ice sheet shows that this ice sheet and glaciers could deliver about 0.2 Tmol Si yr⁻¹ which represents about 60% of the Si riverine inputs to the modern Arctic Sea.

Assuming the same processes were at work the authors estimate that the paleo Greenland ice sheet during the Meltwater Pulse (about 4-5 kyr after the Last Glacial Maximum) might have delivered 5.5 Tmol Si yr⁻¹ to the ocean, thus explaining the high diatom production.

This manuscript is good-science based, well organized, illustrated, and referenced. The data are unique and very innovative. However see below comments in section F.

C-Data/Methodology: generally OK.

-line 340-341: ..."assuming amorphous Si phases dissolved completely within the first hour of extraction". A "supplementary material" section should give examples of kinetic of dissolution during alkaline digestion to support the idea that the "linear regression" method herein used is the most appropriate.

D-Statistics/uncertainties: OK

E-Conclusion: this study shows unique data related to the impact of Greenland ice sheet on the Arctic Sea Si cycle but see below comments(section F).

F-There is a need to strengthen a new version on the manuscript on two points:

(1) although the authors have a world ocean ambition (cf. title) and have a clear vision of the world ocean silica cycle (cf. table 1), there is nothing about the potential impact of the Antarctic ice sheet on the silica cycle which might be potentially more important than that of Greenland. Why there is no word on this key point? If no data is available, the authors should keep modest the title of the manuscript, something like "Ice sheets as a missing source of dissolvable silica to the Arctic ocean" would be more representative of the new findings of this study.

(2) the authors give convincing arguments about (1) the production of amorphous silica (ASi) from glacial weathering processes, and (2) that freshly ground silica particles are much more soluble than classically crushed particles of quartz. As regards the biological utilization of ASi before deposition in fjords and coastal waters, the authors believe this material is highly labile and should be usable by diatoms. Actually, simple in vitro experiments can easily demonstrate the impact of ASi on the growth of diatoms? Were experiments undertaken to support this hypothesis?

G-References: OK

H-Clarity and context: OK

Minor points (corrections):

-line 34:...enhanced production

-line46: ...previously appreciated...

-legend of Figure 1. (b) in east? Greenland

Reviewer #2 (Remarks to the Author):

Congratulations on a well-written manuscript! I found this to be a compelling and interesting contribution examining an overlooked mechanism by which glaciers impact geochemical cycles. For many years, I've argued that sediments, which are widely dispersed and, in the case of eolian materials in particular, are not confined to the boundaries of a watershed, extend the geochemical influence of glaciers. The notion presented here, that a large fraction of the particulate load from ice sheets consists of reactive silica fits squarely in that context. The arguments within the manuscript are well-constructed, with appropriate cautionary statements on the hazards of extrapolating from one glacier to all of Greenland and to global significance. My suggestions are minor, and are aimed at improving the manuscript clarity.

1. I find abbreviations to be an impediment to reading. Some abbreviations are so widely used that they now serve as well or better than the unabbreviated term. For instance GrIS for Greenland Ice Sheet is an effective term because of its widespread use. However, I repeatedly tripped over ASi as an abbreviation for "amorphous particulate silica". It would help readability to spell this one out.

2. Define the term "reactive silica". The working definition here seems to be DSi plus amorphous particulate silica, but when I did a google search on reactive silica, I find it defined as dissolved silica. "Unreactive silica" (the complement to "reactive silica") consists of colloidal silica and polymerized silica. The definition of reactive silica I found on the web does not seem to include amorphous particulate silica. Please define your terms, perhaps with a box, or something else to make it handy for readers to refer to.

3. I found that having methods at the end of the text problematic. I had to jump to the methods in order to understand what each term (dissolved, particulate, amorphous, etc) truly meant. If putting methods at the end is a requirement, so be it. If there is author latitude on this point, I urge putting methods in the standard place, right after the introduction.

4. I was confused by the presentation of amorphous particulate silica as a percentage (line 82 is the first instance). Again, one has to skip to the end of the text to read the methods to learn that amorphous particulate silica is determined from the concentration of dissolved silica liberated by reacting a measured mass of sediment with an alkali. There must be a calculation (undescribed) to convert this concentration back to a %. Further, I cannot follow precisely how the % extractable amorphous particulate silica "equates" (Line 82) to an amorphous particulate silica concentration. I tried reproducing the value of 392 μM for Greenland meltwater from 0.91%, using data for the total SPM load and total discharge from Table 1. I calculated 359 μM , which is 10% lower than the value reported in Table 1 and line 83. Line 83 in the text refers the reader to Figure 2, which does not show this, and does not refer the reader to Table 1, which seems to have the pieces needed to see this "equating". (In fact, looking at Figure 2, snapping a line from 0.9 % ASi seems to equate to ASi of 450 μM . Obviously, the scales don't work this way, but I don't see any way to use Figure 2 to discern the relationship between % amorphous particulate silica and its concentration.

5. In discussion, the saturation indices for quartz and amorphous silica are discussed as "SI-qtz/SI-ASi", yet these SI values are never presented or discussed as an actual ratio.

6. There has been substantial work on effects of suspended sediment delivered to the oceans on seawater chemistry. I pointed to this in a review of the role of glaciers in biogeochemical cycles I wrote in 2007. The work I'm most familiar with comes from Iceland; it seems only fair to recognize this literature. Here are a few papers:

a. Gislason S.R., Oelkers, E.H. and Snorrason Á., (2006). The role of river suspended material in

the global carbon cycle. *Geology* 34, 49-52.

b. Jones M.T., Gislason S.R., Burton K.W., Pearce C.R., Mavromatis V., Pogge von Strandmann P.A.E., Oelkers E.H., (2014). Quantifying the impact of riverine particulate dissolution in seawater on ocean chemistry. *Earth Planet. Sci. Let.* 395, 91-100.

c. Frogner, P, Gislason, SR, Oskarsson, N (2001). Fertilizing potential of volcanic ash in ocean surface water. *Geology* 29, 487-490.

7. Figure 2: please define dw (axis label).

8. Figure 3: caption (lines 667-668) is unclear, and appears to have grammar problems. Explain spectra across bottom of each panel (and increase font size on peak labels).

9. Figure 5: Why not reverse the x-axis on the plot so that the spatial distribution of the sampling points on the map is aligned with the distribution of points on the plot? Obviously the plot axis is not distance, and yet there is a strong correspondence between position and salinity.

Reviewer 1

Reviewer 1 presents a very positive review of the manuscript. They made some excellent suggestions, which we hope to have addressed in the revised manuscript.

Main comments:

- 1) *line 340-341: ..."assuming amorphous Si phases dissolved completely within the first hour of extraction". A "supplementary material" section should give examples of kinetic of dissolution during alkaline digestion to support the idea that the "linear regression" method herein used is the most appropriate.*

This is a widely used method for the determination of ASi in particulate material, and is based on the principle that amorphous material dissolves at a faster rate than crystalline minerals. The linear dissolution of clay has been well documented in previous studies investigating the use of a weak alkaline leach on particulate material (e.g. the original DeMaster (1981) paper in the reference list), therefore we do not consider that an additional section in supplementary materials is necessary. To provide clarity for the reader we have added appropriate references to this sentence.

- 2) *There is a need to strengthen a new version on the manuscript on two points:*
 - (1) *although the authors have a world ocean ambition (cf. title) and have a clear vision of the world ocean silica cycle (cf. table 1), there is nothing about the potential impact of the Antarctic ice sheet on the silica cycle which might be potentially more important than that of Greenland. Why there is no word on this key point? If no data is available, the authors should keep modest the title of the manuscript, something like "Ice sheets as a missing source of dissolvable silica to the Arctic ocean" would be more representative of the new findings of this study.*

We have now added a paragraph on the potential for Antarctic Ice Sheet input of reactive silica into the Southern Ocean (lines 259-286 in the untracked version/lines 297-330 in the tracked version). We show that Antarctic Ice Sheet reactive silica flux is likely to be lower than that of the Greenland Ice Sheet, but still significant. However, it must be noted that little data currently exists and these estimates have high uncertainties – we have noted this in the text. Please note we also considered palaeo ice sheet inputs during meltwater pulse events, which are likely to have been significant in the global silica cycle.

3) *the authors give convincing arguments about (1) the production of amorphous silica (ASi) from glacial weathering processes, and (2) that freshly ground silica particles are much more soluble than classically crushed particles of quartz. As regards the biological utilization of ASi before deposition in fjords and coastal waters, the authors believe this material is highly labile and should be usable by diatoms. Actually, simple in vitro experiments can easily demonstrate the impact of ASi on the growth of diatoms? Were experiments undertaken to support this hypothesis?*

We noted above that to perform in vitro experiments robustly with diatom cultures would require a significant investment of time and resources, and is outside the remit of this manuscript. We did perform a simple seawater leaching experiment with treated (0.1M Na₂CO₃, and hence any ASi removed) and untreated glacial SPM to demonstrate the release of bioavailable DSi from glacial debris. Please see lines 195-209 (229-243 in the tracked version) and Fig. 5. This experiment demonstrated the relatively rapid dissolution of ASi in seawater, (especially compared to the ASi-free sediments), and release of silicic acid, DSi, from ASi, which is bioavailable to marine diatoms.

Minor comments:

-line 34:...enhanced production

Corrected

-line46: ...previously appreciated...

Corrected

-legend of Figure 1. (b) in east? Greenland

Corrected

Reviewer 2

Reviewer 2 gave a very positive and encouraging review of our manuscript. Although they have no major suggestions, there are some minor comments that we hope to have addressed below.

Minor comments:

1) *I find abbreviations to be an impediment to reading. Some abbreviations are so widely used that they now serve as well or better than the unabbreviated term. For instance GrIS*

for Greenland Ice Sheet is an effective term because of its widespread use. However, I repeatedly tripped over ASi as an abbreviation for "amorphous particulate silica". It would help readability to spell this one out.

We only use five abbreviations throughout the manuscript for regularly used terms – DSi, ASi, GrIS, AIS and SPM. ASi is a commonly used term for amorphous silica in literature (see Conley, 1998, Treguer and De La Rocha, 2013, Frings et al., 2014 and Frings et al., 2016 in the reference list, for example). We therefore feel we are justified to use this as an abbreviation in this instance.

2) *Define the term "reactive silica". The working definition here seems to be DSi plus amorphous particulate silica, but when I did a google search on reactive silica, I find it defined as dissolved silica. "Unreactive silica" (the complement to "reactive silica") consists of colloidal silica and polymerized silica. The definition of reactive silica I found on the web does not seem to include amorphous particulate silica. Please define your terms, perhaps with a box, or something else to make it handy for readers to refer to. "Reactive silica" is a term we use to collectively describe dissolved and dissolvable amorphous particulate silica in the manuscript – i.e. silica that is already in dissolved species, or is dissolvable – and is there not classical terminology. To avoid confusion, we have now removed its use in the manuscript, referring instead to "dissolved and amorphous silica" in the text.*

3) *I found that having methods at the end of the text problematic. I had to jump to the methods in order to understand what each term (dissolved, particulate, amorphous, etc) truly meant. If putting methods at the end is a requirement, so be it. If there is author latitude on this point, I urge putting methods in the standard place, right after the introduction.*

This is in compliance with the journal's style.

4) *I was confused by the presentation of amorphous particulate silica as a percentage (line 82 is the first instance). Again, one has to skip to the end of the text to read the methods to learn that amorphous particulate silica is determined from the concentration of dissolved silica liberated by reacting a measured mass of sediment with an alkali. There must be a calculation (undescribed) to convert this concentration back to a %. Further, I cannot follow precisely how the % extractable amorphous particulate silica "equates"*

(Line 82) to an amorphous particulate silica concentration. I tried reproducing the value of 392 μM for Greenland meltwater from 0.91%, using data for the total SPM load and total discharge from Table 1. I calculated 359 μM , which is 10% lower than the value reported in Table 1 and line 83. Line 83 in the text refers the reader to Figure 2, which does not show this, and does not refer the reader to Table 1, which seems to have the pieces needed to see this "equating". (In fact, looking at Figure 2, snapping a line from 0.9 % ASi seems to equate to ASi of 450 μM . Obviously, the scales don't work this way, but I don't see any way to use Figure 2 to discern the relationship between % amorphous particulate silica and its concentration.

We use two terms to describe ASi, as the reviewer highlights. The first is the % of the dry weight of the sediment (i.e. how much of the sediment dry weight is ASi). We use the %ASi and the suspended sediment concentration at the time point of sampling to calculate the concentration of ASi per litre (in μM), as it is often expressed like this in the literature (e.g. see Conley, 1998; or Frings et al., 2016). The mean % and μM concentrations are derived from the means of the 25 samples we analysed.

We reference Fig. 2 on line 84 (line 93 in the tracked version) to point the reader toward the time series of ASi (expressed as both %ASi recorded and the respective concentration in μM), which contains the data from all the samples.

The two scales on Fig. 2 for ASi as %dw and in μM are meant to be read independently with the respective data points.

We have added reference to Table 1 on line 84 (line 93 in the tracked version) in the revised manuscript as suggested. Additionally, we have added a short paragraph to the methodology to describe how %ASi and the concentration of ASi (in μM) are calculated.

- 5) *In discussion, the saturation indices for quartz and amorphous silica are discussed as "SI-qtz/SI-ASi", yet these SI values are never presented or discussed as an actual ratio. This was not meant to be expressed as a ratio. We have changed the text to make this clearer.*

- 6) *There has been substantial work on effects of suspended sediment delivered to the oceans on seawater chemistry. I pointed to this in a review of the role of glaciers in biogeochemical cycles I wrote in 2007. The work I'm most familiar with comes from Iceland; it seems only fair to recognize this literature. Here are a few papers:
a. Gislason S.R., Oelkers, E.H. and Snorrason Á., (2006). The role of river suspended*

material in the global carbon cycle. Geology 34, 49-52.

b. Jones M.T., Gislason S.R., Burton K.W., Pearce C.R., Mavromatis V., Pogge von Strandmann P.A.E., Oelkers E.H., (2014). Quantifying the impact of riverine particulate dissolution in seawater on ocean chemistry. Earth Planet. Sci. Let. 395, 91-100.

c. Frogner, P, Gislason, SR, Oskarsson, N (2001). Fertilizing potential of volcanic ash in ocean surface water. Geology 29, 487-490

We thank the reviewer for pointing this out, and have included some additional references from this group in our revised manuscript.

7) *Figure 2: please define dw (axis label).*

Done.

8) *Figure 3: caption (lines 667-668) is unclear, and appears to have grammar problems. Explain spectra across bottom of each panel (and increase font size on peak labels).*

We have rephrased this caption and increased the font size of the peak labels.

9) *Figure 5: Why not reverse the x-axis on the plot so that the spatial distribution of the sampling points on the map is aligned with the distribution of points on the plot? Obviously the plot axis is not distance, and yet there is a strong correspondence between position and salinity.*

We thank the reviewer for their suggestion, and have changed the figure as suggested.

REVIEWERS' COMMENTS:

Reviewer #1 (Remarks to the Author):

The authors correctly addressed most of my points. The revised version of manuscript is clearly stronger than the initial one. But it is not yet mature for publication.

Point 1-Analytical procedure: alkaline dissolution of siliceous particulate material (amorphous + clays). The authors are quite lucky to get clays/pore crystalline materials that release DSi at a constant rate, as simple as in DeMaster (1981). However, in numerous coastal environments this is not the case. See for instance Ragueneau et al. 2005.

Point 2: "world ambition". Yes, the added paragraph showing data from Antarctica is very useful, definitely. As the authors point out there is still a lot uncertainty as regards the "subglacial meltwater discharge and DSi concentrations as well as no data on ASi for SPM or iceberg hosted sediments."

Point 3: The need for additional in vitro experiments demonstrating the biological utilization of ASi before deposition in fjords and near coastal regions:
The added paragraph is useful to show the lability of ASi (and thus its potential bioavailability). The reported experiment show that the amorphous material is able to dissolve in seawater, at least a significant part of it, in a few months. However, as pointed out by the authors "there is considerable uncertainty around the biological utilization of ASi before deposition in fjords and near coastal regions". In other words if the reported in vitro experiment is clearly useful to show the lability of ASi it does not address the biological utilization issue and the above sentence ("there is ...utilization") should be modified.

So all in all, this manuscript provides unique data which supports the idea that polar ice sheets can provide significant inputs of silicic acid and dissolvable amorphous silica to coastal waters of Greenland and probably to the Antarctic ocean.

Three comments on the revised version:

1-Without doubt the above message is an important message for marine biogeochemists. However, it is based on data acquired in one site of Greenland. Numerous extrapolations are made to go from one site to the whole Greenland system.

2-The total discharge due to the polar ice sheet of Greenland + Antarctica is estimated at about 0.3 Tmol Si yr⁻¹, which is only 3% of the total DSi inputs to the world ocean (Treguer & De La Rocha, 2013; Frings et al. 2016). So the title "ice sheets as a missing source of silica to the world's ocean" is inappropriate. "Ice sheets as a missing source of dissolved silica to the polar oceans" would be more appropriate.

3-The authors made a rough calculation to estimate the dissolved and amorphous inputs from the paleo ice sheets. The DSi+ASi paleo flux is compared to the modern riverine DSi+ASi flux. It should be compared to the paleo riverine flux.

Minor comment:

-line 12: amorphous

Reviewer #2 (Remarks to the Author):

Thank you to the authors for their clarifications of the few queries I had in the original manuscript.
I have no objections to acceptance and publication of this work.

Reviewer 1

Reviewer 1 acknowledges that we address the majority of their initial suggestions. We hope to have addressed their final comments below and in the revised manuscript.

Comments

1) *Analytical procedure: alkaline dissolution of siliceous particulate material (amorphous + clays). The authors are quite lucky to get clays/pore crystalline materials that release DSi at a constant rate, as simple as in DeMaster (1981). However, in numerous coastal environments this is not the case. See for instance Ragueneau et al. 2005.*

The 0.1M Na₂CO₃ leach tested is a well-characterised weak alkaline digest, as described in the manuscript and by other researchers. Some ASi may contain Al (see EDS spectra from HR-TEM analysis of amorphous material for example), therefore it is not appropriate to use an Al correction in terrestrial SPM. The extraction in Ragueneau et al. (2005) paper is a more aggressive 0.2 M NaOH solution at 100°C. It was not considered that some of the lithogenic material that dissolved in a non-linear manner was inorganic ASi. We would argue that some of the initial non-linear dissolution the authors observed may have been ASi.

2) *“world ambition”. Yes, the added paragraph showing data from Antarctica is very useful, definitely. As the authors point out there is still a lot uncertainty as regards the “subglacial meltwater discharge and DSi concentrations as well as no data on ASi for SPM or iceberg hosted sediments.”*

Please see responses to point 4 below.

3) *The need for additional in vitro experiments demonstrating the biological utilization of ASi before deposition in fjords and near coastal regions:*

The added paragraph is useful to show the lability of ASi (and thus its potential bioavailability). The reported experiment show that the amorphous material is able to dissolve in seawater, at least a significant part of it, in a few months. However, as pointed out by the authors “there is considerable uncertainty around the biological utilization of ASi before deposition in fjords and near coastal regions”. In other words if the reported in vitro experiment is clearly useful to show the lability of ASi it does not address the biological utilization issue and the above sentence (“there is ...utilization”) should be modified.

We have now changed this sentence to reflect the reviewers concerns.

4) *Without doubt the above message is an important message for marine biogeochemists. However, it is based on data acquired in one site of Greenland. Numerous extrapolations are made to go from one site to the whole Greenland system.*

We thank the reviewer for suggesting that this is an important message for marine biogeochemists. We agree that our flux estimates are uncertain as they are based on just three sites (one for glacial meltwater runoff, and two for iceberg discharge). We have highlighted in the manuscript (lines 250-252) that more data is needed to constrain these fluxes, and that uncertainties exist in our estimates. Fieldwork of this nature is logistically complex and expensive, therefore these estimates are made to the best of our knowledge. As detailed in point 5, we have toned down the title of the manuscript as suggested.

5) *The total discharge due to the polar ice sheet of Greenland + Antarctica is estimated at about 0.3 Tmol Si yr⁻¹, which is only 3% of the total DSi inputs to the world ocean (Treguer & De La Rocha, 2013; Frings et al. 2016). So the title “ice sheets as a missing source of silica to the world’s ocean” is inappropriate. “Ice sheets as a missing source of dissolved silica to the polar oceans” would be more appropriate.*

We have changed the title to “Ice sheets as a missing source of silica to the polar oceans” to reflect the reviewers concerns.

6) *The authors made a rough calculation to estimate the dissolved and amorphous inputs from the paleo ice sheets. The DSi+ASi paleo flux is compared to the modern riverine DSi+ASi flux. It should be compared to the paleo riverine flux.*

We agree with the author that this would be most ideal, however palaeo river discharge estimates are extremely uncertain. Milliman and Farnsworth (2013) estimate that riverine discharge was at least 20-25% lower from during the Last Glacial Maximum compared to present day. We use this figure to calculate approximate DSi/ASi fluxes from palaeo rivers in our manuscript (lines 312-316). This demonstrates that palaeo ice sheet fluxes may have been equal to non-glacial riverine fluxes.

Minor comment:

-line 12: amorphous

Changed